# Potential of Lycii Radicis Cortex as an Ameliorative Agent for Skeletal Muscle Atrophy

**DOI:** 10.3390/ph17040462

**Published:** 2024-04-04

**Authors:** Rak Ho Son, Myeong Il Kim, Hye Mi Kim, Shuo Guo, Do Hyun Lee, Gyu Min Lim, Seong-Min Kim, Jae-Yong Kim, Chul Young Kim

**Affiliations:** 1College of Pharmacy and Institute of Pharmaceutical Science and Technology, Hanyang University, 55 Hanyangdaehak-ro, Ansan 15588, Republic of Korea; sonnaco@huons.com (R.H.S.); hyemi586@gmail.com (H.M.K.); guoshuo8080@hanyang.ac.kr (S.G.); do247@naver.com (D.H.L.); ossha5gm@gmail.com (G.M.L.); 2R&D Center, Huons Co., Ltd., 55 Hanyangdaehak-ro, Ansan 15588, Republic of Korea; kimmi@huons.com; 3Medical Device Development Center, Daegu-Gyeongbuk Medical Innovation Foundation (DGMIF), 88 Dongnae-ro, Daegu 41061, Republic of Korea; ksm8873@kmedihub.re.kr

**Keywords:** Lycii Radicis Cortex, muscle atrophy, C2C12 myotubes, dexamethasone, protein degradation, protein synthesis

## Abstract

Lycii Radicis Cortex (LRC) is a traditional medicine in East Asia with various beneficial effects, including antioxidant, anti-inflammatory, anti-tumor, anti-diabetic, and anti-depressant properties. However, its potential effects on skeletal muscle atrophy have not been studied. In this study, the protective effects of LRC extract (LRCE) on dexamethasone (DEX)-induced muscle atrophy were investigated in C2C12 myotubes and mice. We evaluated the effect of LRCE on improving muscle atrophy using a variety of methods, including immunofluorescence staining, quantitative polymerase chain reaction (qPCR), Western blot, measurements of oxidative stress, apoptosis, ATP levels, and muscle tissue analysis. The results showed that LRCE improved myotube diameter, fusion index, superoxide dismutase (SOD) activity, mitochondrial content, ATP levels, expression of myogenin and myosin heavy chain (MHC), and reduced reactive oxygen species (ROS) production in dexamethasone-induced C2C12 myotubes. LRCE also enhanced protein synthesis and reduced protein degradation in the myotubes. In mice treated with DEX, LRCE restored calf thickness, decreased mRNA levels of muscle-specific RING finger protein 1 (MuRF1) and atrogin-1, and increased insulin-like growth factor 1 (IGF-1) mRNA level. Moreover, LRCE also repaired gastrocnemius muscle atrophy caused by DEX. Although human studies are not available, various preclinical studies have identified potential protective effects of LRCE against muscle atrophy, suggesting that it could be utilized in the prevention and treatment of muscle atrophy.

## 1. Introduction

The skeletal muscle, constituting around 40% of body weight, plays a pivotal in the whole body’s energy metabolism through elaborate energy production and consumption processes [1]. It fulfills several vital functions, such as maintaining postural and balance, respiratory mechanics, insulin-stimulated glucose uptake and storage, and regeneration after injury [2]. However, both qualitative and quantitative abnormalities in skeletal muscle have been associated with various diseases, including diabetes, cancer, heart failure, and chronic renal failure [3,4]. 

Muscle atrophy is a chronic, progressive muscle disease characterized by excessive protein breakdown compared to synthesis, leading to loss of muscle mass and weakness. It occurs due to various factors such as aging, immobility, disease, injury, genetics, denervation, starvation, and long-term drug therapy. It can lead to decreased physical function, poor quality of life, and an increased risk of falls and fractures [2,5,6,7,8]. Moreover, muscle atrophy is a common feature in numerous chronic diseases such as cancer, heart failure, and chronic obstructive pulmonary disease, exacerbating morbidity and mortality rates associated with these conditions [9]. In recent times, muscle atrophy has emerged as a significant issue that not only places an extra economic burden on the healthcare system, but also poses a global social challenge due to the increased human lifespan resulting from advancements in medicine [10]. However, effective pharmacological therapies have yet to be developed. Therefore, understanding the fundamental mechanisms of muscle atrophy and developing effective strategies for prevention or treatment is crucial.

Pathological conditions characterized by muscle atrophy, such as sepsis, cachexia, starvation, and metabolic acidosis, are associated with an increase in circulating glucocorticoid levels [11]. Dexamethasone (DEX), a potent synthetic glucocorticoid, is widely used in clinical practice for its potent anti-inflammatory and anti-shock properties, as well as its ability to protect against autoimmune diseases [12]. However, prolonged or high-dose DEX treatment has been associated with detrimental effects on skeletal muscle, leading to muscle atrophy [13]. Muscle-specific RING finger protein 1 (MuRF1) and atrogin-1/MAFbx, two key muscle-specific ubiquitin E3 ligases, have emerged as critical regulators of muscle protein degradation pathways implicated in dexamethasone-induced muscle atrophy [14]. These enzymes play a crucial role in inducing muscle protein degradation by activating the ubiquitin-proteasome system (UPS), a major system involved in protein degradation [15]. Another molecular mechanism of DEX-induced muscle atrophy involves the Akt/mTOR/p70S6K pathway, which plays a crucial role in regulating anabolic processes in skeletal muscle by promoting protein synthesis and inhibiting protein degradation [16]. DEX has been shown to suppress the Akt/mTOR/p70S6K pathway, resulting in decreased protein synthesis and increased protein degradation, leading to muscle atrophy [17]. In addition, in mice, DEX-induced muscle atrophy results in reduced calf thickness, muscle mass, and muscle size [18]. Therefore, it is important to identify substances and explore strategies to prevent DEX-induced muscle protein degradation and increase muscle protein synthesis. However, no drug therapy is currently available to prevent DEX-induced muscle protein degradation and increase muscle protein synthesis. Recently, nutraceuticals utilizing various natural substances such as proteins, amino acids, peptides, minerals, vitamins, fatty acids, phytochemicals (polyphenols, flavonoids, polysaccharides, alkaloids, triterpenoids), probiotics, and plants have garnered attention for their preventive and therapeutic effects against skeletal muscle atrophy [19].

Lycii Radicis Cortex (LRC), also known as “Jigolpi” in Korea, is the dried root bark of *Lycium chinese* Mill. LRC has been widely used in Eastern Asia as a traditional medicine to treat lung fever, blood fever, night sweats, pneumonia, cough, hematemesis, inflammation, and diabetes mellitus [20]. Previous studies have reported that LRC exhibits various biological and pharmacological activities, such as antioxidant, anti-inflammatory, and anti-tumor, as well as lowering serum glucose [21]. In addition, several in vitro and in vivo studies have demonstrated the anti-diabetic, anti-depressant, osteogenesis-promoting, and multiple myeloma-preventing effects of LRC [22,23,24,25]. Moreover, recent research has reported that LRC effectively restored DEX-induced bone loss by inhibiting osteoblast apoptosis and upregulating osteoblast differentiation-related genes [26]. Although LRC has various biological roles in the human body, its anti-muscle atrophic function has not been investigated so far. Herein, the effects of Lycii Radicis Cortex extract (LRCE) on DEX-induced muscle atrophy in C2C12 myotubes and mice was investigated for the first time.

## 2. Results

### 2.1. Effects of LRCE on Viability and Atrophy in DEX-Treated C2C12 Myotubes

Through comparison with the literature [27] and HPLC analysis, the main peak in Figure 1A has been identified as kukoamines A and B, the primary component of the LRC extract (LRCE), which demonstrates the suitability of the utilized LRCE in the experiment. To assess the cytotoxicity of LRCE, fully differentiated C2C12 cells were treated with LRCE in various concentrations (5, 25, 50, and 100 µg/mL) for 24 h at 37 °C. The results showed that LRCE at all concentrations tested did not exhibit any toxicity in C2C12 myotubes (Figure 1B). In addition, to determine the protective effects of LRCE against DEX-induced cytotoxicity, the myotubes were incubated with LRCE (5, 25, 50, and 100 µg/mL) and 10 µM DEX for 24 h at 37 °C [28]. DEX-treated myotubes exhibited a 36% reduction in viability compared to control, whereas co-treatment with 100 µg/mL LRCE and 10 µM DEX resulted in a recovery of cell viability (Figure 1C). In particular, co-treatment of 100 µg/mL LRCE with DEX significantly increased the viability compared with DEX-treated C2C12 myotubes (Figure 1C). Furthermore, to assess the reliable phenotype of C2C12 myotubes, immunostaining was performed for myosin heavy chain (MHC), a muscle-specific protein (Figure 1D). DEX-treated myotubes showed significant decreases in diameter (Figure 1D,E), MHC-positive area (Figure 1D,F), and fusion index (Figure 1D,G) compared to the control group. However, treatment with both 50 and 100 µg/mL of LRCE markedly prevented the DEX-induced reductions in MHC-positive area, diameter, and fusion index capacity in C2C12 myotubes (Figure 1D–G), respectively.

### 2.2. Effects of LRCE on the Expression of Muscle Atrophy-Related Factors in DEX-Treated C2C12 Myotubes

DEX treatment significantly suppressed the mRNA expression of myogenin, a muscle differentiation factor, while increasing the mRNA expression of myostatin, MuRF1, and atrogin-1, which are markers of atrophy and protein degradation (Figure 2A). However, treatment with 100 µg/mL LRCE markedly reduced the decrease in myogenin mRNA level caused by DEX (Figure 2A). Furthermore, co-treatment of C2C12 myotubes with LRCE (50 and 100 µg/mL) and DEX resulted in lower mRNA levels of myostatin, MuRF1, and atrogin-1 compared to DEX-treated myotubes (Figure 2A). In addition, compared to the control, DEX treatment significantly reduced MHC protein expression and increased MuRF1 and atrogin-1 protein expression in C2C12 myotubes (Figure 2B–E). In contrast, LRCE treatment significantly increased the expression of MHC and decreased the expression of MuRF1 in DEX-treated C2C12 myotubes (Figure 2B–D).

### 2.3. Effects of LRCE on the Akt/mTOR/p70S6K Pathway in DEX-Treated C2C12 Myotubes

To assess the effect of LRCE on the muscle protein synthesis pathway against DEX treatment in C2C12 myotubes, the phosphorylation levels of Akt, mTOR, and p70S6K were examined via Western blot analysis (Figure 3A). The DEX-treated C2C12 myotubes displayed a significant reduction in the phosphorylation of Akt, mTOR, and p70S6K compared to the control group (Figure 3). Notably, treatment with LRCE (50 and 100 µg/mL) effectively reversed the DEX-induced decrease in the phosphorylation levels of Akt, mTOR, and p70S6K. In fact, the phosphorylation levels of the observed LRCE treatment surpassed those of the untreated control group (Figure 3).

### 2.4. Effects of LRCE on the SOD Activity and ROS Production in DEX-Treated C2C12 Myotubes

In C2C12 myotubes, treatment with 10 µM DEX significantly reduced SOD activity and increased ROS production (Figure 4). However, treatment with LRCE (50 and 100 µg/mL) markedly increased SOD activity, with levels similar to or higher than the control group (Figure 4A). LRCE treatment also reduced the increase in ROS levels induced by DEX in C2C12 myotubes (Figure 4B). In particular, the 50 µg/mL LRCE-treated group showed similar results to the control group, and the 100 µg/mL LRCE-treated group even showed a lower level of ROS production compared to the control group (Figure 4B).

### 2.5. Effects of LRCE on the Expression of BAX and Bcl-2 in DEX-Treated C2C12 Myotubes

To investigate the effect of LRCE on DEX-induced apoptosis in C2C12 myotubes, we examined the expression of BAX and Bcl-2, which are well-known apoptosis-related proteins. DEX treatment significantly increased BAX expression and decreased Bcl-2 expression compared to the control group (Figure 5). However, in the LRCE-treated group, BAX expression was markedly reduced and Bcl-2 expression level was significantly restored compared to the DEX-treated group (Figure 5). Notably, the expression levels of BAX and Bcl-2 in the co-treated group with DEX and LRCE were similar to those of the control group (Figure 5).

### 2.6. Effects of LRCE on Mitochondrial Content and ATP Production in DEX-Treated C2C12 Myotubes

To investigate the effect of LRCE on the quantitative and qualitative improvement of mitochondria in DEX-treated C2C12 myotubes, we used MitoTracker Deep Red staining and measured ATP levels. Treatment with DEX caused a significant decrease in mitochondrial content and ATP levels in C2C12 myotubes compared to the control group (Figure 6). However, C2C12 myotubes treated with DEX and 50 or 100 µg/mL of LRCE significantly recovered the decrease in mitochondrial content and ATP production induced by DEX, respectively (Figure 6).

### 2.7. Effects of LRCE on Body Weight, Calf Thickness, and Gastrocnemius Muscle in DEX-Treated Mice

The body weight and calf thickness of mice injected with DEX (1 mg/kg) significantly decreased compared to the control group. While LRCE treatment did not impact the body weight reduction induced by DEX, it attenuated the decrease in calf thickness induced by DEX (Figure 7A,B). In addition, the mRNA levels of atrogin-1 and MuRF1 in gastrocnemius muscle markedly increased with DEX treatment, whereas these mRNA levels were reduced by LRCE treatment compared to the DEX-treated group (Figure 7C,D). The IGF-1 mRNA level was not affected by DEX, and when LRCE and DEX were co-treated, the IGF-1 mRNA level was higher than that of the control group (Figure 7E). In particular, a significant increase was observed with 100 mg/kg LRCE treatment. Furthermore, DEX reduced the myofiber cross-sectional area of the gastrocnemius muscle compared to the control (Figure 7F,G). The reduction in the gastrocnemius muscle fiber cross-sectional area by DEX was significantly recovered by treatment with LRCE at 50 and 100 mg/kg (Figure 7F,G). Surprisingly, despite DEX treatment, the LRCE-treated groups showed a similar or higher area than the control group (Figure 7F,G).

## 3. Discussion

In recent years, there has been a surge in interest toward developing strategies to prevent or reverse muscle atrophy, particularly focusing on exploring natural extracts or compounds with anti-atrophic properties [29]. Plants have gained global attention as alternative treatments for various diseases due to their affordability, minimal side effects, potent activity, and diverse structures [30]. Recently, there has been increasing attention on food ingredients, plant extracts, and their active compounds as substances promoting muscle health and preventing muscle wasting [31]. In addition, our previous studies demonstrated that natural products such as *Justicia procumbens* L., *Salvia plebeia* R.Br., and rosmarinic acid prevent dexamethasone (DEX)-induced muscle atrophy in C2C12 myotubes [28,32].

LRC has a variety of pharmacological activities and has been widely used in East Asia as a traditional medicine to treat numerous diseases. Kukoamine B (KB), the main constituent of LRC (Figure 1A), is a member of the spermine alkaloid (SA) family and has two dihydrocaffeoyl moieties and one spermine unit [33]. KB has been found to possess the antioxidant and anti-inflammatory properties of common SAs [34,35], and recent studies have demonstrated its ability to protect human keratinocyte HaCaT cells by inhibiting covalent modification by trans-2-nonenal [27]. However, despite these diverse activities, there has been no prior report on the effects of LRC on DEX-induced muscle atrophy.

Numerous previous investigations have demonstrated that DEX-induced muscle atrophy models induce features such as decreased C2C12 myotubes viability, diameter, MHC expression, and fusion index [36,37]. In the present study, LRCE ameliorated the inhibition of DEX-induced death of C2C12 and the decrease in diameter and fusion index (Figure 1C,E,G). These results are consistent with previous findings that treatment with various natural products such as quercetin, myricanol, and fucoxanthin ameliorates DEX-induced muscle atrophy symptoms [37,38,39]. Taken together, these results suggest that LRCE inhibits DEX-induced cytotoxicity and C2C12 myofiber morphological changes.

Muscle atrophy arises from heightened protein breakdown and diminished protein synthesis. Thus, the modulation of protein synthesis and degradation stands as a crucial strategy in managing muscle atrophy [32,40]. Therefore, it is important to increase the expression of myogenin and myosin heavy chain (MHC), which are necessary for muscle cell development [41,42], while inhibiting the expression of myostatin, a marker for inhibiting muscle protein synthesis, and MuRF1 and atrogin-1, markers for protein degradation [43,44]. Black soybean extract (BS Ext) and daidzein had a positive effect on myoblast differentiation and myotube growth by increasing myoD, myogenin, and MHC expression, and prevented DEX-induced myotube growth inhibition [45]. The purified phosphate dried extract (BST204) prevented DEX-induced muscle atrophy by upregulating myotube formation through increasing myogenin and MHC expression and reducing muscle protein degradation through inhibiting the expression of MuRF and atrogin-1 [46]. In addition, natural product-derived substances such as ginsenoside Rg1 and *Justicia procumbens* L. have also shown preventive effects on DEX-induced decreased myotube formation and increased muscle protein degradation in myoblasts [17,32]. In this study, compared to the DEX group, treatment with LCRE resulted in the upregulation of myogenin mRNA levels and MHC protein, while myostatin, murf1, and atrogin-1 mRNA levels, as well as MuRF1 protein, were downregulated (Figure 2). These results indicate that LCRE is an effective substance in improving DEX-induced muscle atrophy.

The Akt/mTOR/p70S6K signaling pathway plays a crucial role in regulating protein synthesis and cellular growth across various tissues, particularly in skeletal muscle. During muscle synthesis, there is an increase in the phosphorylation of Akt and its downstream target mTOR/p70S6K, whereas this phosphorylation decreases during muscle atrophy [16,47]. *Pyropia yezoensis* peptide (PYP15) demonstrated an inhibition of the DEX-induced downregulation of the Akt/mTOR/p70S6K pathway in C2C12 myotubes [48], and monotropin (MON) ameliorated muscle atrophy by regulating catabolic states via the Akt/mTOR/FOXO3a signaling pathway in mice and C2C12 myotubes [49]. Additionally, recent research has indicated that *Salvia plebeia* R.Br. and rosmarinic acid significantly reverses the DEX-induced decrease in Akt/mTOR/p70S6K pathway activity [28]. LRCE was found to enhance the phosphorylation of Akt, mTOR, and p70S6K, which had been reduced by DEX treatment in C2C12 myotubes (Figure 3). LRCE activates the Akt/mTOR/p70S6K signaling pathway, consequently promoting muscle protein synthesis, suggesting its potential to alleviate DEX-induced muscle atrophy.

The SOD enzyme assumes a crucial role in safeguarding cells against oxidative stress by eliminating ROS and curbing their accumulation [50]. Excessive ROS production in myotubes can suppress Akt/mTOR signaling and muscle protein synthesis, leading to accelerated protein degradation [51]. Furthermore, heightened ROS generation stands as a characteristic feature of skeletal muscle atrophy induced by DEX or fasting [52,53]. Previous studies have reported that *Valeriana fauriei* inhibits the increase in ROS and reduction in SOD activity induced by DEX in C2C12 myotubes, suggesting its effectiveness in improving muscle atrophy [36]. These results align with our recent research findings (Figure 4), indicating that LRCE has beneficial effects on muscle atrophy by controlling oxidative stress induced by DEX.

The intimate link between apoptosis and skeletal muscle atrophy has been demonstrated, with older animals showing higher levels of apoptosis compared to younger animals [54]. BAX and Bcl-2, pivotal members of the Bcl-2 protein family, regulate apoptosis [55]. BAX, acting as a pro-apoptotic protein, mediates cell death through mitochondrial permeability transition induction, while Bcl-2, an anti-apoptotic protein, suppresses apoptosis by inhibiting BAX activity [56]. Several studies have shown increased apoptosis in DEX-treated C2C12 myotubes or mice, with co-treatment with natural compounds like quercetin, myricanol, fucoxanthin, and 20(*S*)-ginseonside-Rg3 preventing apoptosis by reducing BAX expression and increasing Bcl-2 expression [37,38,39,57]. Our current study confirmed these findings by observing increased BAX expression and decreased Bcl-2 expression in DEX-treated C2C12 myotubes, both of which were significantly inhibited by LRCE treatment (Figure 5). These results suggest that LRCE may prevent DEX-induced muscle atrophy by reducing apoptosis through the suppression of BAX expression and promotion of Bcl-2 expression.

Mitochondrial dysfunction contributes to skeletal muscle wasting observed in various muscle atrophy models such as diabetes, obesity, disuse, and aging [58,59,60]. In disuse muscle atrophy, mitochondrial content decreased, leading to a reduction in ATP production and contributing to the loss of muscle mass and strength [61,62]. In addition, DEX treatment reduces mitochondrial content and ATP production rate in C2C12 myotubes compared to the control [63]. Thus, maintaining normal mitochondrial content and ATP production capacity is crucial for preserving both mitochondrial health and muscle function [64,65]. LRCE restored mitochondrial content and the ATP levels reduced by DEX in C2C12 myotubes (Figure 6). Therefore, our findings suggest that LRCE might prevent muscle atrophy by mitigating mitochondrial dysfunction in DEX-treated C2C12 myotubes.

In mice, DEX-induced muscle atrophy manifests as reduced body weight, calf thickness, muscle size, and protein content [66]. Our study found that while LRCE did not affect DEX-induced weight loss, the calf thickness of mice supplemented with 100 mg/kg LRCE significantly increased compared to those treated with 1 mg/kg DEX alone (Figure 7A,B). DEX preferentially affects muscles containing type IIb fast-twitch fibers, such as the gastrocnemius, which is primarily composed of these fibers [67,68]. Therefore, we analyzed mRNA and cross-sectional area changes in gastrocnemius muscles. We found that DEX treatment significantly increased the mRNA expression of atrogin-1 and MuRF1, markers of muscle catabolism, while LRCE treatment alleviated these increases (Figure 7C,D). Interestingly, DEX treatment did not affect insulin-like growth factor 1 (IGF-1) mRNA expression, a major anabolic growth factor stimulating the PI3K/Akt signaling pathway. However, IGF-1 mRNA levels in mice treated with DEX and LRCE at 100 mg/kg were more than three-fold higher compared to the control or DEX-treated groups (Figure 7E). In addition, LRCE treatment significantly restored the cross-sectional area of gastrocnemius muscles in DEX-treated mice (Figure 7F,G). These results suggest that LRCE can prevent muscle atrophy by inhibiting catabolic activity and promoting anabolic activity within muscle tissue. Overall, this study suggests the potential of LRCE as a functional material for preventing muscle atrophy by targeting multiple pathways involved in muscle protein synthesis and degradation, oxidative stress, apoptosis, and mitochondrial function, and to the best of our knowledge, it is the first report on the effect of LRCE on ameliorating DEX-induced muscle atrophy in vitro and in vivo.

## 4. Materials and Methods

### 4.1. Plant Materials

The LRC plant materials used in this study were purchased from Songlim Pharm. Co., Ltd. (Seoul, Republic of Korea) and collected in July 2020 in Henan Province, China. They were identified by the corresponding author (CY Kim) and stored at room temperature until used in the study. A voucher was deposited at the Pharmacognosy Laboratory of the College of Pharmacy, Hanyang University (specimen No. HYUP-LR-001). The amount of 150 kg of dried LRC was extracted two times with 2100 L of 50% ethanol at a temperature of 85 ± 5 °C for 6 h, additionally containing 1.6% citric acid. Afterward, the extract was evaporated in vacuo at 65 °C. Finally, the resulting concentrate was blended with maltodextrin DE20 and spray dried in a spray dryer to produce approximately 30 Kg of LRCE, which was used in the experiments.

### 4.2. Sample Preparation and Optimization of HPLC Analysis

LRCE powder was dissolved at a concentration of 500 mg/100 mL in 10% methanol (1% acetic acid). Kukoamines A and B (ChemFaces, Wuhan, Hubei, China) were dissolved in 10% methanol (1% acetic acid) and used for analysis. The analytical method was optimized by adjusting the chromatographic parameters such as solvent, column, gradient range of elution, flow rate, column temperature, mobile phase, and detection wavelength. HPLC analysis was performed using an Agilent Technologies 1200 system (Agilent Technologies, Santa Clara, CA, USA) equipped with an automatic injector, a column oven, and a DAD detector. The chromatography was performed using a Zorbax SB-aq (4.6 × 250 mm, 5 µm, Agilent Technologies) column at 30 °C and detected at 280 nm of UV wavelength. The mobile phase consisted of acetonitrile (0.1% trifluoroacetic acid, solvent A) and water (0.1% trifluoroacetic acid, solvent B) in a gradient mode: 0–5 min, 10% A; 5–25 min, 10–15% A; 25–35 min, 15–100% A; 35–40 min, 100% A. The flow rate was 1 mL/min, and the injection volume was 10 μL. The diode array detector employed a UV spectrum over a range of 210 to 400 nm. For a quantitative analysis of the kukoamines A and B (KA and KB) in the extract, the calibration curves for the KA and KB were obtained by plotting the peak area versus the concentration for each analyte by least-square regression analysis. Calibration equation was obtained using five levels of concentrations ranging from 0.1 to 1.0 mg/mL.

### 4.3. C2C12 Cell Culture and Differentiation

The C2C12 mouse skeletal muscle cell line was purchased from American Type Culture Collection (ATCC, Manassas, VA, USA) and maintained in high-glucose (25 mM) DMEM (Gibco Corporation, New York, NY, USA) supplemented with 10% FBS, 100 U/mL penicillin, and 100 µg/mL streptomycin at 37 °C in a humidified 5% CO_2_ atmosphere. Myoblasts were seeded onto 6-well plates (1.2 × 10^5^ cells/well) for Western blot and qPCR analysis, 12-well plates (6 × 10^4^ cells/well) for immunostaining, and 48-well plates (1 × 10^4^ cells/well) for cell viability measurements. When C2C12 myoblasts reached approximately 90% confluence, the growth medium was replaced with a differentiation medium (high-glucose DMEM containing 2% HS, 100 U/mL penicillin, and 100 µg/mL streptomycin) for 6 days to induce myotube differentiation [32].

### 4.4. Treatment of LRCE and DEX

After differentiation, myotubes were subdivided into four groups as follows: (1) the control group, in which cells were incubated in DMEM supplement HS and 0.1% DMSO (Sigma-Aldrich, St. Louis, MO, USA) for 24 h; (2) the DEX (Sigma-Aldrich, MO, USA)-treated group, in which cells were treated with 10 µM of DEX for 24 h; and (3–4) co-treatment groups of LRCE with DEX, in which cells were treated with DEX in the presence of LRCE (50 and 100 µg/mL) for 24 h. Subsequently, all groups were harvested for the next experiments.

### 4.5. Cell Viability Assay

Cell viability analysis followed established protocols [32]. Briefly, C2C12 myoblasts were seeded in 48-well plates at a density of 1 × 10^4^ cells per well for 48 h and were fully differentiated into C2C12 myotubes for 6 days. Subsequently, C2C12 myotubes were treated with 10 μM DEX in the presence or absence of LRCE (50 and 100 μg/mL) for 24 h, respectively. Afterward, 20 µL of CCK-8 reagent was added to each well and followed incubation for 4 h at 37 °C. Absorbance was then measured at 450 nm using an EnSpire Multimode Plate Reader (PerkinElmer, Waltham, MA, USA).

### 4.6. Myosin Heavy Chain (MHC) Immunofluorescence Staining

Immunofluorescence staining was performed as described previously [32]. Myotubes were initially fixed using a solution containing 4% paraformaldehyde for 10 min at room temperature. Subsequently, they underwent three washes with phosphate-buffered saline (PBS). To facilitate permeabilization, myotubes were treated with 0.1% Triton X-100 (Sigma-Aldrich, MO, USA) for 20 min, followed by blocking with 3% BSA (Biosesang, Seongnam, Korea) for 1 h at room temperature. Following blocking, the myotubes were incubated with anti-MHC (1:300, Santa Cruz, Dallas, TX, USA) overnight at 4 °C. After washing three times with 0.1% phosphate-buffered saline-Tween 20 (PBST), the myotubes were incubated with a secondary antibody conjugated with Alexa Fluor 488 (1:500, Invitrogen, Waltham, MA, USA) at 37 °C for 1 h. Nuclei were subsequently counterstained with 10 µM of Hoechst 33342 (Sigma-Aldrich, MO, USA). The myotubes’ diameters were measured for a total of 100 myotubes from at least 5 different fields. For the MHC-positive area analysis, 5 randomly selected fields were counted from three independent experiments in each group. The fusion index was calculated with the equation as follows: (number of nuclei inside MHC-positive myotubes in 5 fields)/(total number of nuclei present in 5 fields). Measurements of myotube diameters, MHC stained area, and fusion index were conducted and analyzed using Image J software (1.48 version).

### 4.7. Quantitative Real-Time PCR

Total RNA was extracted from either C2C12 myotubes or gastrocnemius muscle tissue of mice using TRIzol reagent (Invitrogen, Waltham, MA, USA). The concentration and purity of each RNA sample were assessed by determining the absorbance ratio at 260 nm and 280 nm (A260/A280). For myotubes, cDNA synthesis was performed by the reverse transcription of 1 μg total RNA using the ReverTra Ace™ qPCR RT master mix (TOYOBO, Osaka, Japan). In the case of gastrocnemius muscle tissue, cDNA was synthesized from 5 μg total RNA using the GoScript™ Reverse Transcription Mix, Oligo(dT) kit (Promega, WI, USA). Real-time quantitative PCR (qPCR) was subsequently conducted using specific mouse primers and THUNDERBIRD™ SYBR^®^ qPCR mix (TOYOBO, Osaka, Japan). The amplified products from real-time PCR were quantified using the comparative cycle threshold (Ct) method. Each gene was normalized to the expression level of GAPDH, and the analysis was conducted using the CFX Maestro V. 2.3 Software (Bio-Rad, Hercules, CA, USA). Primer information is provided in Appendix A.

### 4.8. Westerm Blot Analysis

Western blot was performed as described previously [32]. C2C12 myotubes were washed twice with ice-cold PBS and lysed with cold RIPA lysis buffer (Invitrogen, NY, USA) containing 1% protease/phosphatase inhibitor cocktails (Invitrogen, Waltham, MA, USA) on ice for 20 min. The protein concentration of each sample was determined using the Pierce BCA Protein Assay Kit (Thermo Fisher Scientific, MA, USA). Subsequently, equal amounts of protein (20 µg) were loaded onto either 8% or 12% SDS polyacrylamide gels and separated via electrophoresis, followed by transfer onto PVDF membranes (Merck, Darmstadt, Germany) for 1 h. The membranes were then blocked with a 3% BSA solution for 2 h at room temperature. Following blocking, the membranes were probed with specific primary antibodies against MHC, mTOR, MuRF1, atrogin-1, BAX, and Bcl-2 (1:1000, Santacruz, TX, USA); β-actin and GAPDH (1:5000, Santacruz, TX, USA); and Akt, p-Akt, p-mTOR, p70S6K, and p-p70S6K (1:1000, Cell Signaling Technology, MA, USA) overnight at 4 °C. The next day, the membranes were washed three times with 0.2% PBST and subsequently incubated with the appropriate anti-mouse (1:5000, Santacruz, TX, USA) or anti-rabbit (1:5000, Santacruz, TX, USA) IgG HRP-linked antibodies for 2 h at room temperature. Subsequently, the membranes were washed three times with 0.2% PBST. The protein was detected by ECL Western Blotting Substrate (Thermo Fisher Scientific, MA, USA). Images were captured using ChemiDocTM XRS + Imaging System (Bio-Rad, CA, USA), and band densitometry analysis was performed using Image J software (1.48 version).

### 4.9. Measurement of SOD Activity

SOD activity was assessed according to the instructions of the SOD assay kit (Dojindo, Kumamoto, Japan). Briefly, 20 µL of each cell lysate was transferred into a 96-well plate, followed by the addition of 200 µL of WST working solution and 20 µL of enzyme working solution. The mixture was then incubated at 37 °C for 20 min, and SOD activity was subsequently measured at 450 nm using an EnSpire Multimode Plate Reader (PerkinElmer, Waltham, MA, USA).

### 4.10. Measurement of Intracellular ROS

Intracellular ROS levels were determined using a fluorescent probe (CM-H2DCFDA) (Invitrogen Waltham, MA, USA). The myotubes were exposed with 10 µM DEX in the presence or absence of LRCE (50 and 100 µg/mL) for 24 h. Following treatment, myotubes were washed with PBS and incubated with 10 μM of CM-H2DCFDA solution for 15 min in the dark at 37 °C. After incubation, myotubes were washed twice with culture media, followed by the addition of 200 μL of media to each well. Fluorescence intensity was measured using a fluorescent multi-plate reader at excitation (495 nm)/emission (525 nm) wavelength [30].

### 4.11. Mitochondria Staining

Fully differentiated myotubes were treated with 10 μM DEX in the presence or absence of LRCE (50 and 100 µg/mL) for 24 h. Subsequently, myotubes were stained with 500 nM Mito Tracker Deep Red (Invitrogen, Waltham, MA, USA) for 30 min at 37 °C. After staining, the myotubes were washed with PBS, fixed with 4% paraformaldehyde, and then subjected to washes three times with PBS. Images were captured using a fluorescence microscope (JuLI^TM^ stage, NanoEntek, Seoul, Republic of Korea) [38]. Red fluorescence-positive cells were counted in at least 5 randomly selected fields from each well in three independent experiments, and the fluorescence intensity was analyzed using Image J software (1.48 version).

### 4.12. Determination of ATP Level

ATP levels within the myotubes were measured using a luminescent ATP detection assay kit according to the manufacturer’s instructions (Cayman Chemical, Ann Arbor, MI, USA). Briefly, the myotubes were washed with cold PBS and then homogenized in the ATP detection sample buffer. The homogenized solution was centrifuged for 10 min at 13,000× *g*, and 10 µL of the supernatant was transferred into each well of a white 96-well plate. Then, 100 µL of ATP reaction mix solution was added to each well and incubated at room temperature for 20 min. After the reaction, ATP levels were measured using an EnSpire Multimode Plate Reader at 560 nm (PerkinElmer, Waltham, MA, USA).

### 4.13. Mouse Preparation and Treatment

All animal experiments were approved by the Animal Ethical and Welfare Committee of the Korea Basic Science Institute (No. KBSI-20-31), and all procedures were carried out following approved guidelines and regulations. Male ICR mice (7–8 weeks old, 20–25 g) were obtained from Raonbio Inc. (Yongin, Republic of Korea) and maintained under standard conditions (specific pathogen-free) with air filtration, at a temperature of 22 ± 1 °C with 12 h light/dark cycles. The mice were fed a regular chow diet and given water ad libitum. Prior to inducing muscle atrophy, LCR extracts (50 and 100 mg/kg/day) were administered to the mice via oral gavage for 14 days. After 14 days, muscle atrophy was induced by a single injection of DEX (1 mg/kg/day, s.c.) for 10 days. The mice were divided into four groups (*n* = 8): (1) the control group, in which mice were injected with 30% PEG (polyethylene glycol); (2) the DEX group, in which mice were injected with DEX; and (3–4) co-treatment LRCE with DEX groups, in which mice were administered with LRCE (50 and 100 mg/kg/day) for 14 days and then injected with DEX. The body weight and calf thickness of each mouse were recorded daily. Calf thickness was measured consistently for each individual using an electronic digital caliper at the mid-belly, the thickest part of the gastrocnemius muscle of the left hindlimb leg. The gastrocnemius muscles were dissected, weighed quickly using an analytical balance (Mettler Toledo, Columbus, OH, USA), and frozen in liquid nitrogen [38].

### 4.14. Histological Analysis of Gastrocnemius Muscle

The gastrocnemius muscle was fixed in 10% formalin overnight and then embedded in paraffin. The prepared paraffin block was cut into 4 μm sections for H&E staining to analyze histological changes. The muscle fiber cross-sectional diameter was estimated using the CellSens image analysis system (Olympus, Tokyo, Japan). Using a microscope with a 20× objective, a total of 20 randomly selected views were taken from each slide containing a cross-section of the muscle. The diameter of all muscle fibers in each view was automatically calculated by the CellSens image analysis system. The muscle fiber diameter of each group was expressed as a percentage of the control group [38].

### 4.15. Statistical Analyses

All data are presented as the mean ± standard deviation (SD) for at least three independent experiments. Statistical significance was evaluated and determined by one/two-way analysis of variance (ANOVA) using GraphPad Prism 5.0 (GraphPad Software Inc, La Jolla, CA, USA), followed by Tukey’s post hoc test. A *p*-value of less than 0.05 was considered statistically significant.

## 5. Conclusions

In the present study, LRCE alleviated DEX-induced muscle atrophy in C2C12 myotubes by inhibiting protein degradation, ROS production, apoptosis, and mitochondria dysfunction. Additionally, it improved SOD activity and muscle protein synthesis. Moreover, LRCE reduced muscle tissue damage in mice affected by DEX-induced muscle atrophy. Given the current lack of effective treatments for skeletal muscle atrophy, these findings demonstrate the potential of LRCE as a nutritional supplement to improve muscle atrophy and as a therapeutic agent for prevention and treatment. However, more detailed research, including human studies, is necessary to fully understand LRCE’s anti-atrophy effects and its potential clinical applications.

## Figures and Tables

**Figure 1 pharmaceuticals-17-00462-f001:**
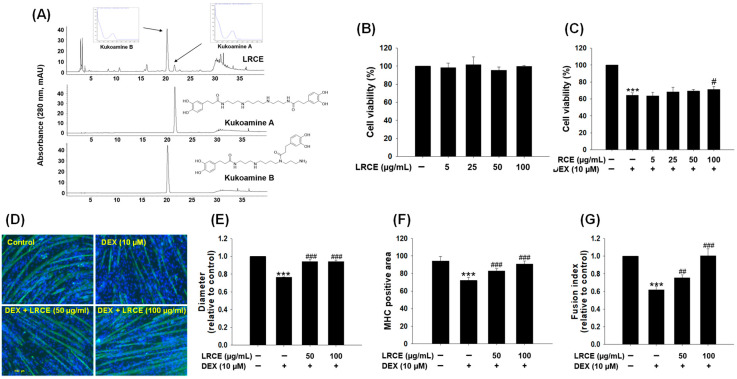
Effects of LRCE on viability and morphology in DEX-treated C2C12 myotubes. (**A**) HPLC-UV chromatogram of LRCE (10 mg/mL), KB (0.4 mg/mL), and KA (0.4 mg/mL). (**B**) The cell viability was measured using the CCK-8 assay. C2C12 myotubes were treated with various concentrations of LRCE (5, 25, 50, and 100 μg/mL) for 24 h. (**C**) C2C12 myotubes were treated with DEX (10 μM) in the presence or absence of LRCE (5, 25, 50, and 100 μg/mL) for 24 h. (**D**) C2C12 myotubes were stained with MHC (green) and DAPI (blue), and representative photographs were observed under a fluorescent microscope (scale bar = 250 μm). (**E**) Relative changes of diameters in myotubes, (**F**) MHC-stained area, and (**G**) fusion index were measured from randomly selected fields and quantified using the Image J program (1.48 version). These results are presented as means ± SD of three independent experiments: *** *p* < 0.001 vs. control; ^#^ *p* < 0.05, ^##^ *p* < 0.01, ^###^ *p* < 0.001 vs. DEX.

**Figure 2 pharmaceuticals-17-00462-f002:**
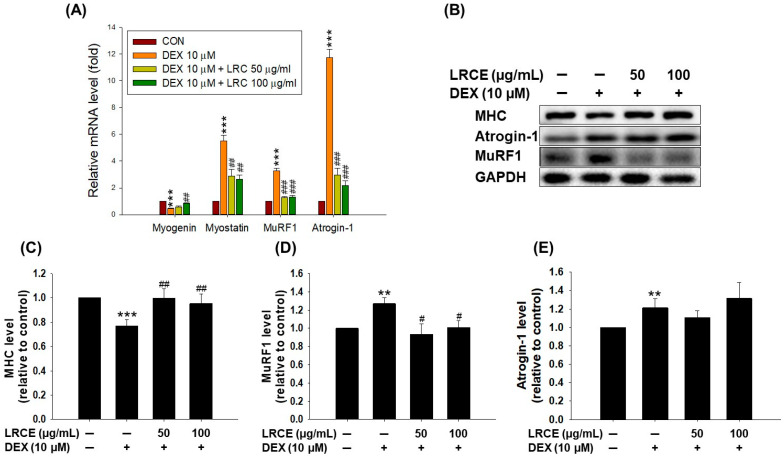
Effects of LRCE on the expression of muscle atrophy-related markers in DEX-treated C2C12 myotubes. (**A**) qRT-PCR analysis for expression of myogenin, myostatin, MuRF1, and atrogin-1 in C2C12 myotubes. (**B**) The expression of MHC, atrogin-1, MuRF1, and GAPDH were analyzed by Western blotting. GAPDH was used as the loading control. (**C**) Quantitative analysis of MHC, (**D**) MuRF1, and (**E**) atrogin-1. These results are presented as means ± SD of three independent experiments: ** *p* < 0.01, *** *p* < 0.001 vs. control; ^#^ *p* < 0.05, ^##^ *p* < 0.01, ^###^ *p* < 0.001 vs. DEX.

**Figure 3 pharmaceuticals-17-00462-f003:**
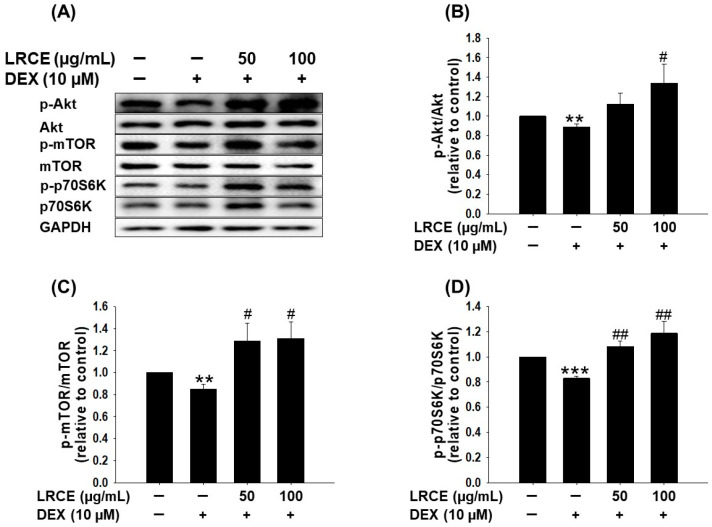
Effects of LRCE on the Akt/mTOR/p70S6K pathway in DEX-mediated atrophy in C2C12 myotubes. (**A**) Western blot of p-Akt, Akt, p-mTOR, mTOR, p-p70S6K, and p70S6K in the C2C12 myotubes treated with 10 μM DEX in the presence or absence of LRCE (50 and 100 μg/mL) for 24 h. (**B**) Quantitative analysis of the p-Akt/Akt. (**C**) Quantitative analysis of the p-mTOR/mTOR. (**D**) Quantitative analysis of the p-p70S6K/p70S6K. These results are presented as the means ± SD of three independent experiments: ** *p* < 0.01, *** *p* < 0.001 vs. control; ^#^ *p* < 0.05, ^##^ *p* < 0.01 vs. DEX.

**Figure 4 pharmaceuticals-17-00462-f004:**
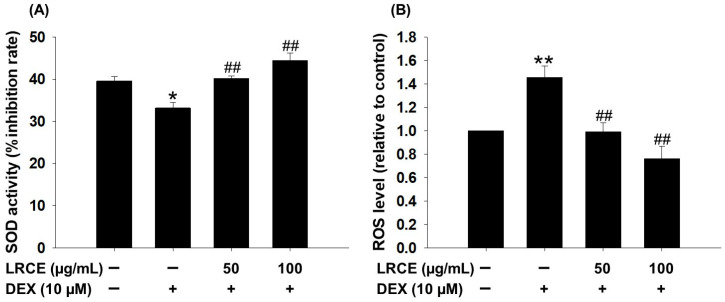
Effects of LRCE on SOD activity and ROS production in atrophy-induced C2C12 by DEX treatment. (**A**) The antioxidant enzyme (SOD) activity. (**B**) The production of cellular ROS analysis using a 2,7-dichlorodihydrofluorescein diacetate (DCFH-DA) dye and measured by a fluorescent multi-plate reader. These results are presented as the means ± SD of three independent experiments: * *p* < 0.05, ** *p* < 0.01 vs. control; ^##^ *p* < 0.01 vs. DEX.

**Figure 5 pharmaceuticals-17-00462-f005:**
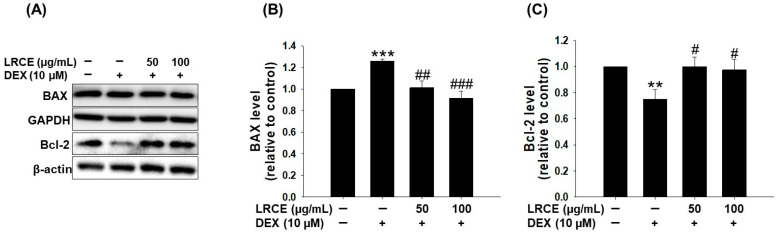
Effects of LRCE on an apoptosis-related protein in atrophy-induced C2C12 by DEX treatment. (**A**) The expression of BAX and Bcl-2 was detected by Western blotting analysis. (**B**) The quantitative of BAX and (**C**) Bcl-2 expression levels. The graph shows a quantitative representation of the levels of protein. These results are presented as the means ± SD of three independent experiments: ** *p* < 0.01, *** *p* < 0.001 vs. control; ^#^ *p* < 0.05, ^##^ *p* < 0.01, ^###^ *p* < 0.001 vs. DEX.

**Figure 6 pharmaceuticals-17-00462-f006:**
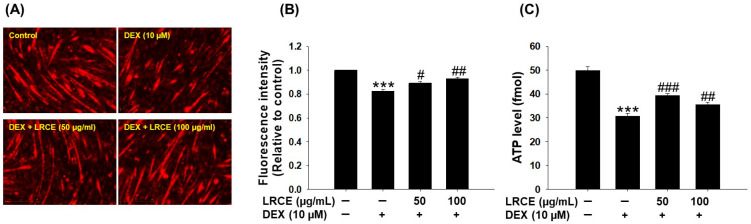
Effects of LRCE on mitochondrial content and ATP levels in DEX-treated C2C12 myotubes. (**A**) Mitochondrial content was determined by MitoTracker Deep Red staining and observed under a fluorescence microscope (Scale bar = 250 μm). (**B**) The graph shows the quantification of the MitoTracker Deep Red-stained area. (**C**) ATP production in the C2C12 myotubes. These results are presented as means ± SD of three independent experiments: *** *p* < 0.001 vs. control; ^#^ *p* < 0.05, ^##^ *p* < 0.01, ^###^ *p* < 0.001 vs. DEX.

**Figure 7 pharmaceuticals-17-00462-f007:**
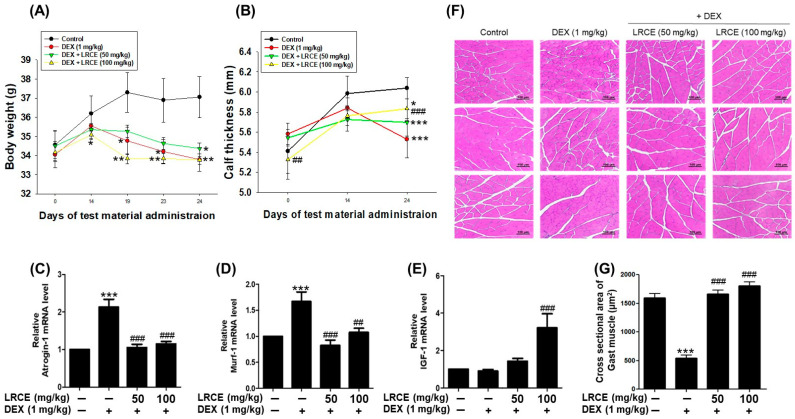
Effects of LRCE on DEX-induced muscle atrophy in mice. (**A**) Body weight. (**B**) Calf thickness. qRT-PCR analysis for expression of (**C**) atrogin-1, (**D**) MuRF1, and (**E**) IGF-1 in gastrocnemius muscle. (**F**) Representative hematoxylin and eosin (H&E) staining of myofiber cross-section of the gastrocnemius muscle (scale bar = 100 μm). (**G**) Quantitative analysis of the cross-sectional area in the gastrocnemius muscle. These results are presented as means ± SD. * *p* < 0.05, ** *p* < 0.01, *** *p* < 0.001 vs. control; ^##^ *p* < 0.01, ^###^ *p* < 0.001 vs. DEX.

## Data Availability

Data will be made available upon request.

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
