# Peer review of "Potential of Lycii Radicis Cortex as an Ameliorative Agent for Skeletal Muscle Atrophy"

_pharmaceuticals, 2024, doi:10.3390/ph17040462_

Round 1

Reviewer 1 Report

Comments and Suggestions for Authors

In the present article, the authors make a very comprehensive evaluation of the effect of a Lycii root extract (LRCE) on dexamethasone-induced muscle atrophy. The formulation of the experiments, controls and results allow us to clearly dissect the pathways that partly explain why protein synthesis is restored and their degradation is reduced. Measurements of ATP, mitochondrial function and oxidative stress markers are also a relevant contribution. Without a doubt, the use of an ad hoc animal model is a very sound contribution of this study and allows us to adequately conclude that this plants have valuable bioactive metabolites. The per se effect of LRCE upon IGF-1 mRNA levels is striking. This event  is related to the activation of the Akt/mTOR pathway. The work is very well designed and easy to read. In my opinion it is an interesting contribution and has merit to be published.

Minor comments

- Although it is true that the observed activity can be attributed to kukoamine B, it would be appreciated to indicate the tentative identity of the peak that appears around 15 min in Figure 1 A. Its concentration is not negligible and therefore it could have some pharmacological effect in conjunction with kukoamine B.

- I recommend including a brief discussion relating the effects observed for kukoamine B and what has been previously reported for spermidine. Discuss the structure-activity relationship between this type of compounds and what advantage, if any, kukoamine B would have over the other smaller amines that seem to share the same mechanism of action.

- Dexamethasone also affects total levels of Foxo3a and its phosphorylated form. This factor plays a key role in atrophy. Why was it not evaluated?

- The conclusion appears somewhat weak for the magnitude of the findings. I suggest strengthening this point.

Comments on the Quality of English Language

Overall, the manuscript requiere minor english editing. 

Reviewer 2 Report

Comments and Suggestions for Authors

Authors evaluated the effect of Lycii Radicis Cortex extract (LRCE) on improving muscle atrophy using a variety of methods, including immunofluorescence staining, qPCR, western blot, measurements of oxidative stress, apoptosis, ATP levels, and muscle tissue analysis. The results showed that LRCE improved myotube diameter, fusion index, SOD activity, mitochondrial content, ATP levels, expression of myogenin and MHC, and reduced ROS production in dexamethasone-induced C2C12 myotubes. LRCE also enhanced Akt/mTOR/p70S6K pathway activation and reduced protein degradation in the myotubes. In mice treated with DEX, LRCE restored calf thickness, decreased mRNA levels of MuRF1 and atrogin-1, and increased IGF-1 mRNA level. Moreover, LRCE also repaired gastrocnemius muscle atrophy caused by DEX. Overall, these findings suggest that LRCE has potential protective effects against muscle atrophy, and could be utilized for the prevention and treatment of muscle atrophy in patients. This study has some deficiencies and cannot be accepted in the current form.

Here are the points:

·       Authors shoud have mentioned in detail how they prepared the extract of LRCE.

·       If the active ingredient is kukoamine B, why authors did not isolate it and did not perform studies with kukoamine B?

·       The HPLC data should be given seperately and the figure is not clear to identify the kukoamine B.

·       Where are the results of the atrophy level of dexamethasone in C2C12 myotubes? What is the concentration of used dexamethasone in C2C12 myotubes? All the further experimets depend on that so authors should clarify.

·       It does not look possible that LRCE can cross the lipophilic barriers to show its effects on atrophy.

·       The introduction and Discussion parts must be enhanced.

·       The Figures are impossible to read. The quality and the size must be increased.

Reviewer 3 Report

Comments and Suggestions for Authors

Dear authors,

The manuscript entitled "Potential of Lycii Radicis Cortex as an ameliorative agent for skeletal muscle atrophy" evaluate the protective effects of LRC extract (LRCE) on dexamethasone (DEX)-induced muscle atrophy were investigated in C2C12 myotubes and mice. It presents scientific relevance for Medicine, Pharmacy, Chemistry and others area. However, you need to change some details/information in the Abstract, Introduction, Material and Methods, results and discussion, and conclusions.

1. Abstract: Adequate! But:

- The abstract is well written, with information of the methods used. There is no numerical data! I suggest inserting the results obtained (numerical data!!!) more relevant.

- I suggest informing the meaning of the acronym "qPCR": Real-time polymerase chain reaction (qPCR). Idem for: SOD, MHC, ROS, MuRF1 and IGF-1.

- The authors wrote: “…Overall, these findings suggest that LRCE has potential protective effects against muscle atrophy, and could be utilized for the prevention and treatment of muscle atrophy in patients.” I suggest reviewing the writing, as to be "used in patients" other in vitro and in vivo tests must be performed!

- At the end, I suggest highlighting the advantages/ disadvantages of the study and methods.

- Keywords: The word "Protein synthesis" is not indicated in the abstract or title. I suggest reviewing!

2. Introduction section:

- Adequate! However, at the end of the introduction, the objective of the manuscript must be indicated. Also, to highlight the "innovative" proposal of the methods, as well as the advantages/disadvantages/limitations of the study.

3. Results section (or “Results and discussion”)

Wouldn't it be more interesting to combine the "results” with the "discussion" to better describe the findings and compare them with other works published in the literature?? I suggest expanding the discussions!

- Page 2, lines 78-80 in “2.1. Effects of LRCE on viability and atrophy in DEX-treated C2C12 myotubes” section: The author wrote “To assess the cytotoxicity of LRCE, fully differentiated C2C12 cells were treated with LRCE in various concentrations (5, 25, 50, and 100 μg/mL)”. How were these concentrations defined? Any protocol from the authors’ research group/laboratory? Or was it based on previous literature/studies? If yes, provide reference! Idem for line 83, DEX (10 μM)!

- I suggest improving the quality of all figures!

- The results are interesting! I suggest discussing of the results obtained by comparing them with the literature!

4. Discussion section (or “Results and discussion”)?

Wouldn't it be more interesting to combine the "results” with the "discussion" to better describe the findings and compare them with other works published in the literature??

- If the authors choose not to combine the results with the discussions, I suggest dividing them into subsections (the same as the results) and discussing each subsection! I suggest expanding the discussions!

- Page 7, lines 236-250: There is no discussion; there is a repetition of the findings, according to the results collected in the figures! I suggest expanding the discussions!

- I suggest, at the end of the "results and discussion", to write a paragraph summarizing the findings and their impacts on the research proposal.

5. Materials and methods section: The methodological proposal is appropriate to the manuscript, but I suggest:

- Page 8, in “4.1. Plant materials” section: What are the conditions for collection/acquisition and storage of samples? What is the time/period (from acquisition to analysis)?

- Page 8, in “4.2 HPLC analysis” section: How were the optimal conditions and mobile phase defined? Any protocol from the authors’ research group/laboratory?  Or was it based on previous literature/studies? Were the HPLC analyzes qualitative or quantitative? If quantitative, I suggest reporting aspects of analytical validation! Have the HPLC methods been validated? What protocol was followed? How were the parameters evaluated: LOD, LOQ, precision, accuracy, robustness and linearity? Which concentration ranges were studied? I suggest detailing the proposed method in more detail...

- For the items (4.3 to 4.12): Please, provide references of methods!

- Page 11, in “4.13. Mouse preparation and treatment” section: The authors wrote “…LCR extracts (50 and 100 mg/kg/day) were administered to the mice via oral gavage for 14 days…”. The authors reported good results at these concentrations! But, were lower concentrations tested?

- Page 11, line 431, in “4.13. Mouse preparation and treatment” section: The authors wrote “…The mice were divided into four groups (n=8): (1) control group, in which mice were injected with 30% PEG…”. What does the acronym PEG mean? Polyethyleneglycol? This is the first time this substance appears in the manuscript!? What is its purpose? What results were obtained in this group? Not shown in the figures!

5. Conclusion: I suggest inserting the results obtained more relevant. I suggest pointing out the main results and disadvantages/limitations of the method and the study!

6. Table and Figures: Adequate. I suggest improving the quality of all figures!

7. Supplementary Materials: Adequate!

8. References: Please, check if the references are in accordance with the journal's rules.

Comments on the Quality of English Language

The language (English) is satisfactory (but, I suggest the final revision)! 

Reviewer 4 Report

Comments and Suggestions for Authors

In the article “Potential of Lycii Radicis Cortex as an ameliorative agent for skeletal muscle atrophy” the authors showed that the extract Lycii Radicis Cortex counteracts the dexamethasone-induced muscle atrophy in c2c12 and in mice. The article is well written and designed, however, in my opinion, before the publication in Pharmaceuticals, the authors should address some modifications.

·         Myosin heavy chain expression should be analysed in gastrocnemius muscles by western blotting;

·         Since in c2c12 they demonstrate that Lycii Radicis Cortex exerts its beneficial effects through Akt/mTOR/p70S6K should be interesting to validate if also in muscle mice the pathway involved are the same.

Round 2

Reviewer 2 Report

Comments and Suggestions for Authors

Authors performed the revision point by point. However, some points are still missing. The HPLC- UV chromatogram of LRCE is impossible to read and is not acceptable in this form. Therefore, authors must change it with a larger and clearer one. They said they increased the quality and size of the Figures as suggested by the reviewer but the figures are the same in the low quality and they are impossible to read. They did not enhance the Introduction. They only added an unnecessary paragraph explaining the skeletal muscle. The iThenticate report is very high, which is not acceptable. Authors should decrease the similarity properly.

Reviewer 3 Report

Comments and Suggestions for Authors

The authors improved the manuscript based on my observations, making it suitable for publication!

Comments on the Quality of English Language

I only suggest a final review!

Reviewer 4 Report

Comments and Suggestions for Authors

In the revised version of the article “Potential of Lycii Radicis Cortex as an ameliorative agent for skeletal muscle atrophy” the authors explain that they should not perform the western blotting analysis required and that the analysis provided are sufficient to demonstrate the beneficial effects of Lycii Radicis Cortex. Thus, the current version of the manuscript in my opinion is suitable for the publication in Pharmaceuticals.
